# Antitumor Activities of tRNA-Derived Fragments and tRNA Halves from Non-pathogenic *Escherichia coli* Strains on Colorectal Cancer and Their Structure-Activity Relationship

Kai-Yue Cao,[a] Yu Pan,[a] Tong-Meng Yan,[a] Peng Tao,[b,c] Yi Xiao,[b,c] Zhi-Hong Jiang[a]

[a]State Key Laboratory of Quality Research in Chinese Medicine, Macau University of Science and Technology, Taipa, Macau SAR, China
[b]School of Physics, Huazhong University of Science and Technology, Wuhan, China
[c]Key Laboratory of Molecular Biophysics of the Ministry of Education, Huazhong University of Science and Technology, Wuhan, China

**ABSTRACT** tRNAs purified from non-pathogenic *Escherichia coli* strains (NPECSs) possess cytotoxic properties on colorectal cancer cells. In the present study, the bioactivity of tRNA halves and tRNA fragments (tRFs) derived from NPECSs are investigated for their anticancer potential. Both the tRNA halves and tRF mimics studied exhibited significant cytotoxicity on colorectal cancer cells, with the latter being more effective, suggesting that tRFs may be important contributors to the bioactivities of tRNAs derived from the gut microbiota. Through high-throughput screening, the EC83 mimic, a double-strand RNA with a 22-nucleotide (nt) 5′-tRF derived from tRNA-Leu(CAA) as an antisense chain, was identified as the one with the highest potency (50% inhibitory concentration [$IC_{50}$] = 52 nM). Structure-activity investigations revealed that 2′-*O*-methylation of the ribose of guanosine (Gm) may enhance the cytotoxic effects of the EC83 mimic via increasing the stability of its tertiary structure, which is consistent with the results of *in vivo* investigations showing that the EC83-M2 mimic (Gm modified) exhibited stronger antitumor activity against both HCT-8 and LoVo xenografts. Consistently, 4-thiouridine modification does not. This provides the first evidence that the bioactivity of tRF mimics would be impacted by chemical modifications. Furthermore, the present study provides the first evidence to suggest that novel tRNA fragments derived from the gut microbiota may possess anticancer properties and have the potential to be potent and selective therapeutic molecules.

**IMPORTANCE** While the gut microbiota has been increasingly recognized to be of vital importance for human health and disease, the current literature shows that there is a lack of attention given to non-pathogenic *Escherichia coli* strains. Moreover, the biological activities of tRNA fragments (tRFs) derived from bacteria have rarely been investigated. The findings from this study revealed tRFs as a new class of bioactive constituents derived from gut microorganisms, suggesting that studies on biological functional molecules in the intestinal microbiota should not neglect tRFs. Research on tRFs would play an important role in the biological research of gut microorganisms, including bacterium-bacterium interactions, the gut-brain axis, and the gut-liver axis, etc. Furthermore, the guidance on the rational design of tRF therapeutics provided in this study indicates that further investigations should pay more attention to these therapeutics from probiotics. The innovative drug research of tRFs as potent druggable RNA molecules derived from intestinal microorganisms would open a new area in biomedical sciences.

**KEYWORDS** tRF, chemical modifications, gut microbiota, antitumor activity

The microbiome in the human gastrointestinal tract, consisting of trillions of microorganisms, comes into being within days after birth (1). Recent studies revealed that the gut microbiota in general provides beneficial effects to the host by aiding gastrointestinal and immune functions (2). While it can promote health, it may also sometimes cause diseases.

Address correspondence to Zhi-Hong Jiang, zhjiang@must.edu.mo.

The authors declare no conflict of interest.

For example, non-pathogenic fatty liver, type 2 diabetes, inflammatory bowel disease, obesity, and colorectal cancer (CRC) have been linked to the microbiota (3). *Escherichia coli* is a Gram-negative gut commensal bacterium found in the colon of mammals and reptiles (4–6). It has become one of the most important model organisms due to its fast growth in chemically defined media *in vitro*. *E. coli* can be categorized into four main phylogenetic groups, A, B1, B2, and D. Pathogenic *E. coli* strains (PECSs) from group B2 have been found to cause gastroenteritis (7), neonatal meningitis (8), hemorrhagic colitis (9), urinary tract infections (10), and Crohn's disease (11), etc. Advanced studies have revealed that PECSs containing the peptide synthetase-polyketide synthetase (PKS) island could induce DNA double-strand breaks in human cell lines, and thus, PECSs may be able to accelerate the development of colorectal cancer (12–14), which is a leading cause of cancer-related deaths worldwide (15, 16). On the other hand, *E. coli* strains of groups A and D are not known to cause diseases and can even be beneficial to their hosts, e.g., by producing vitamin $K_2$ (5, 17, 18). Thus, the functions of these non-pathogenic *E. coli* strains (NPECSs) demand attention from researchers.

In one of our previous studies, tRNA-Val(UAC) and tRNA-Leu(CAG), purified from NPECSs by two-dimensional liquid chromatography, have been shown to exhibit significant cytotoxicity on HCT-8 human colorectal cancer cells (19). As the most abundant class of small RNAs (<200 nucleotides [nt]) (20), tRNAs have been revealed to regulate RNA splicing, RNA translation, and DNA replication (21). It has been reported that endogenous tRNAs may be cleaved by multiple ribonucleases and, under stress conditions, may produce tRNA-derived fragments (tRFs) and tRNA halves (22–25), which have been associated with broad biological functions, such as microRNA (miRNA)-like regulation of protein translation and cellular stress responses (26, 27). In addition, endogenous tRFs can suppress human breast cancer by targeting YBX1, suggesting possible applications as pharmacological agents (28). On the other hand, tRNA halves were found to suppress breast cancer by targeting *FZD3* (29). Therefore, it is reasonable to hypothesize that the cytotoxic effects of tRNA-Val(UAC) and tRNA-Leu(CAG) from NPECSs reported in our previous paper (19) may be mediated by the formation of tRFs and tRNA halves. Therefore, we embarked on a study to evaluate and compare tRNA halves and tRF mimics (double-stranded RNA with tRF as an antisense chain) derived from these two RNAs in terms of their antitumor activities against CRC *in vitro* and *in vivo*. Moreover, the structure-activity relationship of tRF mimics was investigated using chemically modified tRF mimics.

## RESULTS

**tRFs may be important contributors to the bioactivities of tRNAs from NPECSs.** tRNA-Val(UAC) and tRNA-Leu(CAG) were first purified from NPECS total tRNAs (see Fig. S1 in the supplemental material), which were further treated by S1 nuclease to prepare tRNA halves. The results of liquid chromatography-mass spectrometry (LC-MS) analysis and urea-polyacrylamide gel electrophoresis (PAGE) analysis showed that intact tRNAs were successfully cleaved into two half fragments at their anticodon loop (Fig. 1A to C). Cytotoxicity studies, using HCT-8 cells with liposomal transfection at a concentration of 50 nM, showed that intact tRNA-Val(UAC) and tRNA-Leu(CAG) and their tRNA halves and tRF mimics (double-strand RNAs with 22-nt-long 5′- and 3′-tRFs as the antisense chains) significantly decreased the cell viability of HCT-8 cells to various extents, ranging from approximately 36 to 80%, with the exception of the 5′-tRNA half of tRNA-Val(UAC) (Fig. 1D and E). However, the 5′-tRF mimics of both tRNAs are the most effective RNAs.

**5′-tRF-Leu(CAA) with a length of 22 nt exhibits the strongest inhibitory effects on CRC cells.** Double-stranded RNA has been widely employed in investigations on the molecular functions of miRNA or tRF since the double-stranded format is more stable than the single-stranded one (30, 31). A total of 82 tRFs, including 5′-tRFs and 3′-tRFs 22 nt in length derived from 41 tRNAs of NPECSs reported in the MODOMICS database and tRNA db (25) (Table S1), were selected as the antisense strands of tRF mimics in the small interfering RNA (siRNA) form (Table S2) since they are the most abundant types of tRFs (32). Their cytotoxic effects were screened using HCT-8 cells (Fig. 2A). Following an initial screen at 50 nM, a total of 12 tRF mimics that exhibited the greatest effects were selected for

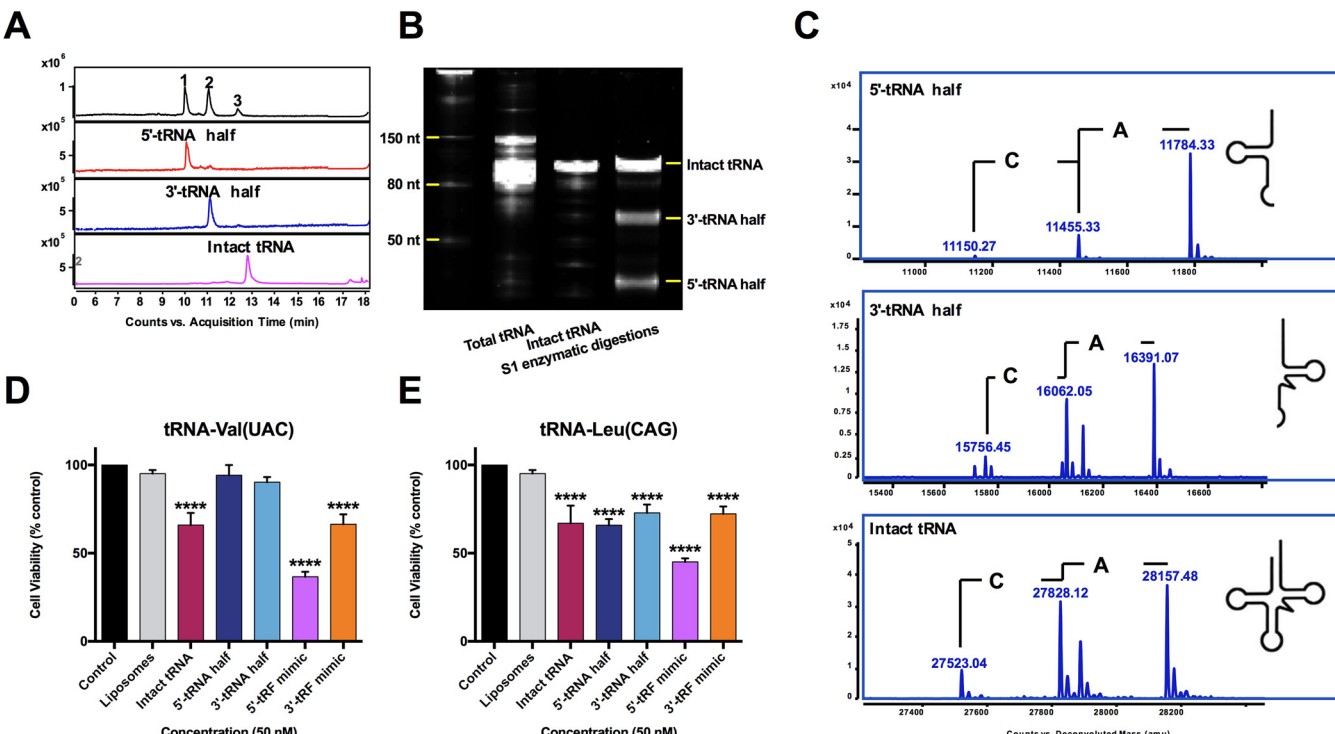

**FIG 1** The cytotoxicity of tRF mimics is significantly stronger than that of tRNA halves on HCT-8 cells. (A) Typical UHPLC chromatogram of 5′- and 3′-tRNA halves of tRNA-Leu(CAG) under UV at 260 nm. (B) Urea-polyacrylamide gel electrophoresis of 5′- and 3′-tRNA halves of tRNA-Leu(CAG) confirming that the purified tRNA was successfully digested into two products. (C) UHPLC-MS analysis of 5′- and 3′-tRNA halves of tRNA-Leu(CAG) showing that the molecular weights of the digestion products were in accordance with the 5′- and 3′-tRNA halves of *E. coli* tRNA. (D) Cytotoxic comparison of tRNA halves, tRF mimics, and individual tRNAs of tRNA-Val(UAC) on HCT-8 cells. (E) Cytotoxic comparison of tRNA halves, tRF mimics, and individual tRNAs of tRNA-Leu(CAG) on HCT-8 cells. Data are shown as the means ± SD from three independent experiments. ****, $P < 0.0001$ (by one-way ANOVA followed by *post hoc* analysis).

detailed analysis. Figure 2B shows the dose-dependent cytotoxic effects of these 12 tRF mimics. The most effective tRF mimic is the EC83 mimic [a double-strand RNA with a 22-nt 5′-tRF derived from tRNA-Leu(CAA) as an antisense chain], with a 50% inhibitory concentration ($IC_{50}$) value of 52.8 nM (Fig. 2C).

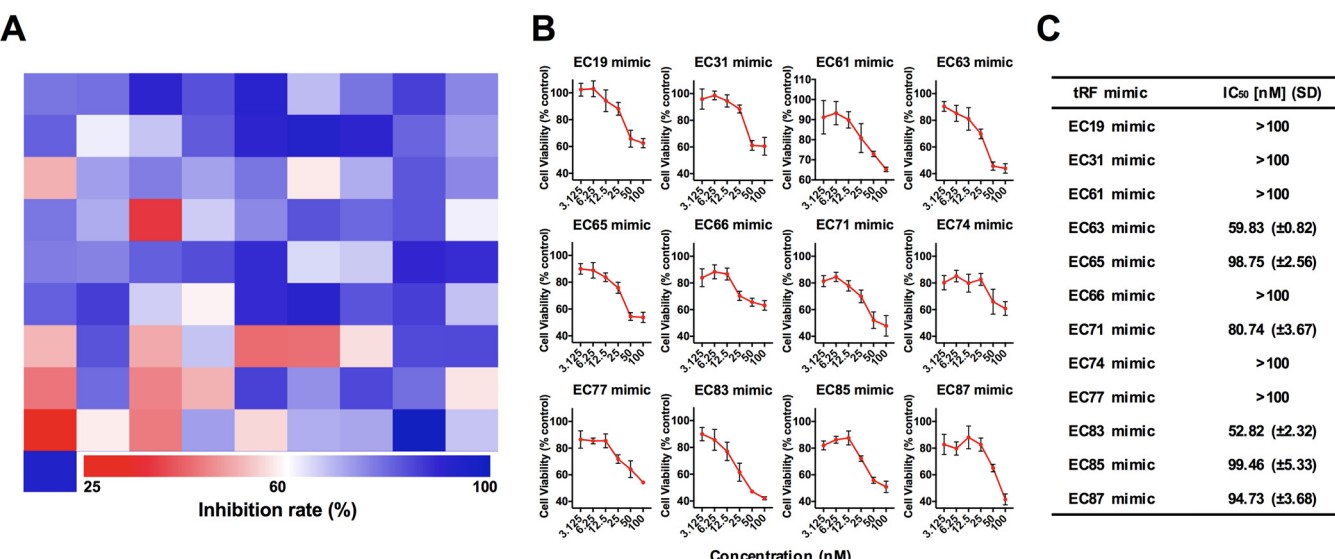

**FIG 2** High-throughput screening hits of EC83 mimics with the strongest cytotoxicity compared to other NPECS tRF mimics on HCT-8 cells. (A) Heatmap of high-throughput screening of 82 tRF mimics with hits of 12 tRF mimics with potency in reducing the cell viability of HCT-8 cells. (B) Dose-dependent investigations of the top 12 hits of tRF mimics. (C) $IC_{50}$ values with standard deviations for the top 12 hits of tRF mimics showing that the EC83 mimic has the strongest cytotoxicity against HCT-8 cells.

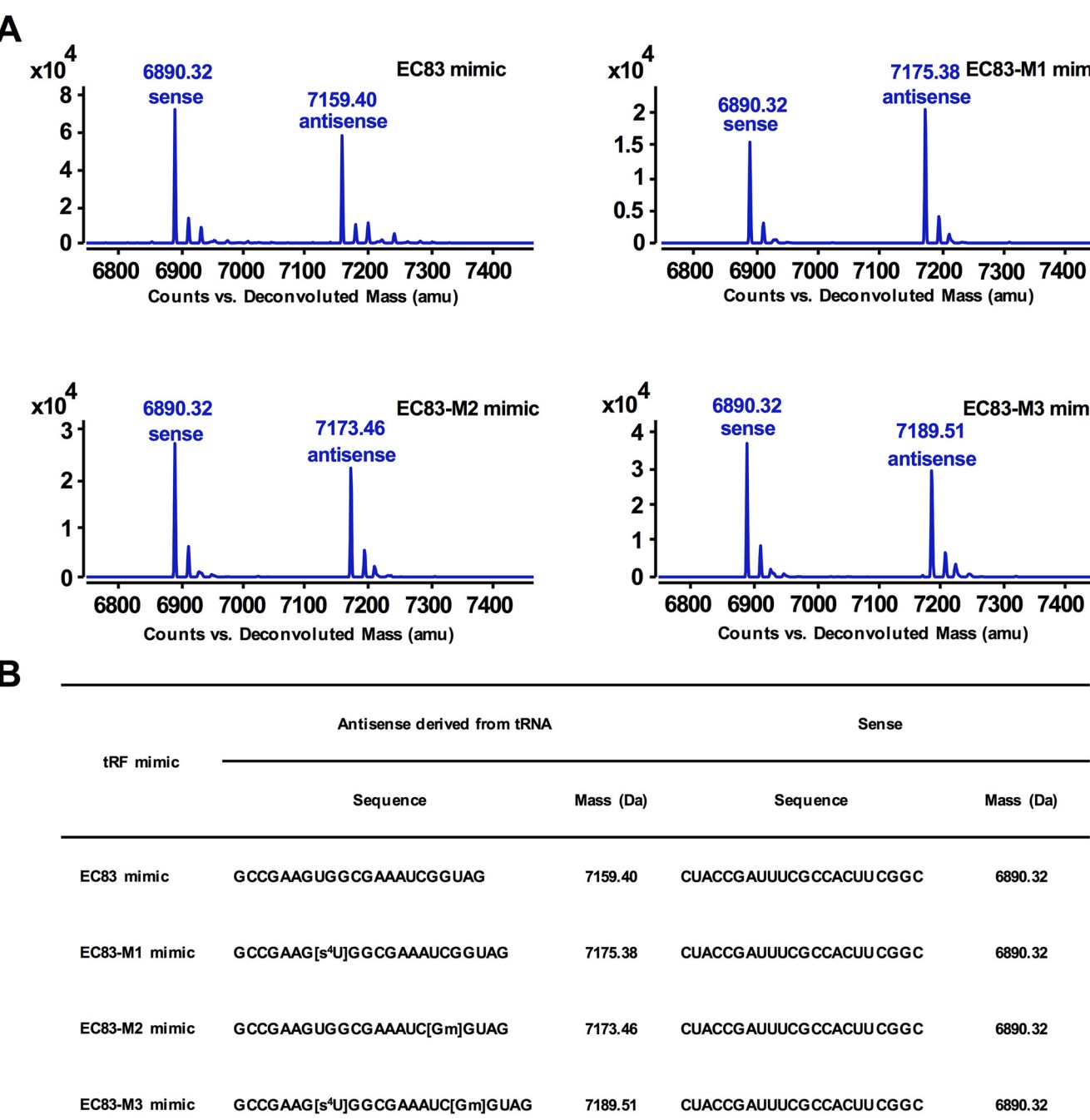

**FIG 3** UHPLC-MS analysis confirmed the accurate molecular weights of the EC83 mimic and its chemically modified derivatives. (A) UHPLC-MS analysis of the EC83 mimic and its modified derivatives confirmed the agreement of their molecular weights with the theoretical values. (B) Sequence information and deconvolution MS of the EC83 mimic and its modified derivatives.

**Gm modification increases the cytotoxicity of EC83 mimics toward CRC cells.** It has been demonstrated that chemical modifications of tRNA can alter the biological functions of tRFs (33). According to the reported modifications of 2′-O-methylguanosine (Gm) and 4-thiouridine (s⁴U) in tRNA-Leu(CAA), three modified derivatives of EC83 were designed and synthesized, namely, the EC83-M1 mimic (s⁴U modified), the EC83-M2 mimic (Gm modified), and the EC83-M3 mimic (both s⁴U and Gm modified). Figure 3A shows the results of the LC-MS analysis, which confirmed the good agreement of their molecular weights with the theoretical values (Fig. 3B) (34). Figure 4A shows that the EC83 mimic and its derivatives decreased the cell viability of colorectal cancer cells (HCT-8 and its fluorouracil [5-FU]-resistant strain and LoVo

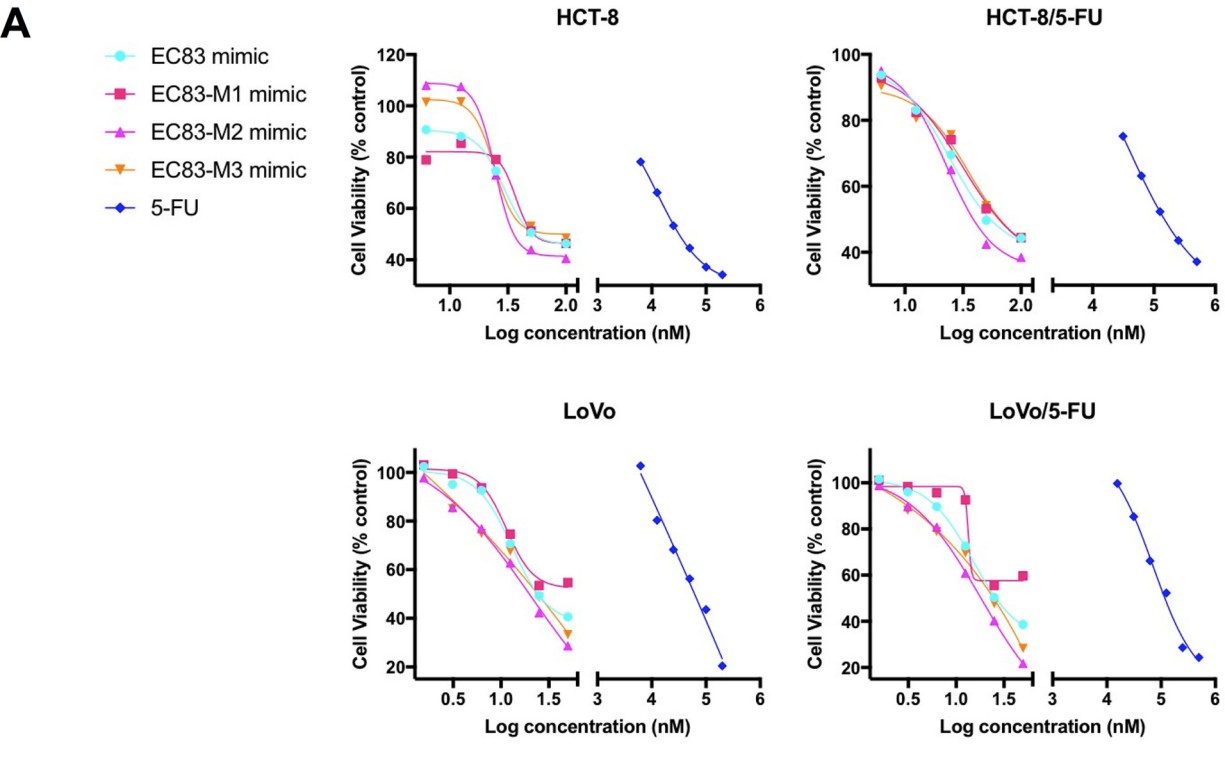

**A**

*Legend:*
- EC83 mimic
- EC83-M1 mimic
- EC83-M2 mimic
- EC83-M3 mimic
- 5-FU

**B**

| Cell line | IC$_{50}$ [nM] (SD) | | | | |
|---|---|---|---|---|---|
| | EC83 mimic | EC83-M1 mimic | EC83-M2 mimic | EC83-M3 mimic | 5-FU |
| HCT-8 | 70.60 (±2.31) | 77.91 (±1.85)[a] | 58.88 (±2.54)[b] | 74.46 (±3.34) | 41400.23 (±30.63) |
| HCT-8/5-FU | 63.43 (±2.12) | 69.62 (±2.84) | 48.82 (±3.21)[b] | 72.24 (±4.56)[a] | 170601.98 (±300.23) |
| LoVo | 28.84 (±2.57) | 43.64 (±4.66)[b] | 19.93 (±1.51)[a] | 24.08 (±3.79) | 63900.38 (±54.96) |
| LoVo/5-FU | 29.48 (±4.32) | 54.23 (±6.51)[c] | 18.27 (±2.82)[a] | 23.03 (±3.16) | 133200.62 (±495.27) |

**FIG 4** Gm modification increases the cytotoxicity of EC83 mimics on CRC cells and their 5-FU-resistant strains, while s⁴U does not. (A) Dose-dependent investigation of EC83 mimics and 5-FU on HCT-8, 5-FU-resistant HCT-8, LoVo, and 5-FU-resistant LoVo cells. (B) IC$_{50}$ values with SD of EC83 mimics and 5-FU demonstrating that the EC83-M2 mimic exhibits stronger cytotoxicity than the EC83 mimic, while the EC83-M1 mimic has weaker cytotoxicity. a, $P < 0.1$; b, $P < 0.05$; c, $P < 0.01$ (by one-way ANOVA followed by *post hoc* analysis).

and its 5-FU-resistant strain). Notably, the EC83-M2 mimic exhibited the strongest cytotoxicity with the lowest IC$_{50}$ value among the four tRF mimics in the four cancer cell lines (Fig. 4B). These results indicated that Gm modification may increase the cytotoxicity of the EC83 mimic, while s⁴U modification does not. In comparison, 5-FU has IC$_{50}$ values in the micromolar concentration range, which is more than 700 times of those of the EC83-M2 mimic, indicating this tRF mimic's extraordinary cytotoxicity toward cancer cells. It is noteworthy that all tRF mimics showed the same potency in the 5-FU-resistant cells as that in the nonresistant cells. Furthermore, three cancer cell lines (the human ovarian cancer cell line A2780, the liver cancer cell line HepG2, and the breast cancer cell line MDA-MB-231) together with the human colon

epithelial cell line HCoEpiC were supplemented in the 3-(4,5-dimethylthiazol-2-yl)-2,5-diphenyl-tetrazolium bromide (MTT) assay as control groups. As shown in Fig. S2A, the results demonstrated that no significant cytotoxicity of the EC83 mimic and its modified derivatives was observed on A2780 cells, but they exhibited strong cytotoxicity on HepG2 and MDA-MB-231 cells, in which the EC83-M2 mimic had a much lower $IC_{50}$ value than those of the other RNAs (Fig. S2B). Meanwhile, no significant cytotoxicity of these tRF mimics was observed on HCoEpiC cells. The above-described results demonstrated that the cytotoxicity of tRF mimics derived from non-pathogenic *Escherichia coli* has cell specificity.

**Gm modification increases the inhibitory effects of EC83 mimics on colony formation and migration of CRC cells.** The effectiveness of the EC83 mimic and its modified derivatives in preventing colony formation in a clonogenic assay was determined on HCT-8 and LoVo cells. The results indicated that all tRF mimics together with 5-FU are very effective in suppressing clonogenic ability in both cell lines. The EC83-M2 mimic (Gm modified) exhibited significantly stronger inhibitory effects on the colony formation of both HCT-8 and LoVo cells than the EC83 mimic, while the EC83-M1 mimic and the EC83-M3 mimic did not exhibit such effect (Fig. 5A).

The migration of cancer cells is important for cancer development. As shown in Fig. 5B, at 24 and 48 h, EC83-M2 mimic-treated HCT-8 cells at 50 nM ($-38.6\% \pm 10.6\%$ and $-48.0\% \pm 5.3\%$, respectively) exhibited a significantly lower wound-healing rate (WHR) than EC83 mimic-treated cells ($4.0\% \pm 2.4\%$ and $2.8\% \pm 5.3\%$), while the EC83-M1 mimic-treated ones had a higher WHR at the same concentration ($19.4\% \pm 2.1\%$ and $11.4\% \pm 2.8\%$). For LoVo cells, EC83-M2 mimic-treated cells at 25 nM ($-4.3\% \pm 4.0\%$ and $-1.9\% \pm 2.7\%$) exhibited a WHR comparable to that with the EC83 mimic ($-2.6\% \pm 5.1\%$ and $-1.6\% \pm 2.1\%$) at 24 and 48 h, while the EC83-M1 mimic-treated ones had a significantly higher WHR at the same concentration ($6.8\% \pm 3.7\%$ and $16.3\% \pm 2.9\%$). Meanwhile, EC83-M3 mimic-treated cells had lower WHRs than those of EC83-M2 mimic-treated cells. As a positive control, 5-FU-treated HCT-8 cells ($11.7\% \pm 4.2\%$ and $13.8\% \pm 2.9\%$) and LoVo cells ($10.1\% \pm 2.6\%$ and $-9.5\% \pm 3.0\%$) at 50 $\mu$M exhibited a low WHR at 24 and 48 h. These results, together with those of the clonogenic assay, demonstrated that Gm modification may increase the cytotoxic effectiveness of the EC83 mimic toward CRC cells, while $s^4U$ modification does not.

**Gm modification increases the inhibitory effects of EC83 mimic toward CRC tumor growth *in vivo*.** To further confirm the antitumor activity of EC83 mimic and EC83-M2 mimic and their structure-activity relationship, a CRC cell (HCT-8 and LoVo) xenograft-bearing nude mouse model was developed for *in vivo* experiments. The results showed that at day 22, compared to the control group, both EC83 mimic and EC83-M2 mimic could significantly suppress the tumor growth rate (Fig. 6A), tumor weight (Fig. 6B), and tumor size (Fig. 6C), while 5-FU was used as a positive control. Interestingly, the EC83-M2 mimic exhibited significantly stronger inhibitory effects than those of EC83 mimic, suggesting that Gm modification might enhance the anticancer activity of EC83 mimics. This is consistent with the *in vitro* results. However, 5-FU was found to significantly reduce the body weight of nude mice at the end of the experiment, while no significant change was observed in the RNA-treated group (Fig. 6D). Meanwhile, hematoxylin and eosin (H&E) staining showed that EC83 mimic and EC83-M2 mimic-treated tumors have less eosinophilic cytoplasm than the control group, indicating their efficacy as tumor suppressants (Fig. 6E). To determine the side effects of the two RNAs, morphological imaging and H&E staining of major organs, including heart, liver, spleen, lung, and kidney, from HCT-8 and LoVo xenograft-bearing nude mice were performed (Fig. S3 and Fig. 6F). The results demonstrated that there are no significant differences in the major organs between the RNA-treated group and the control group, suggesting that RNA treatment is potentially safe.

**Gm modification increases the stability of the EC83 mimic's tertiary structure.** To evaluate the stability of the EC83 mimic and its derived modified mimics at the molecular level, their structures were initially assumed to be the standard A-form double helix from molecular dynamics (MD) simulations. Starting from each of these initial structures, a 100-ns simulation at 300 K in isothermal-isobaric (NPT) ensemble revealed that the EC83 mimic remained as an A-form double helix, while the three modified derivatives transformed into more compact helical conformations rapidly (Fig. 7A). Moreover, as shown in Fig. 7B and C, the fluctuation of the

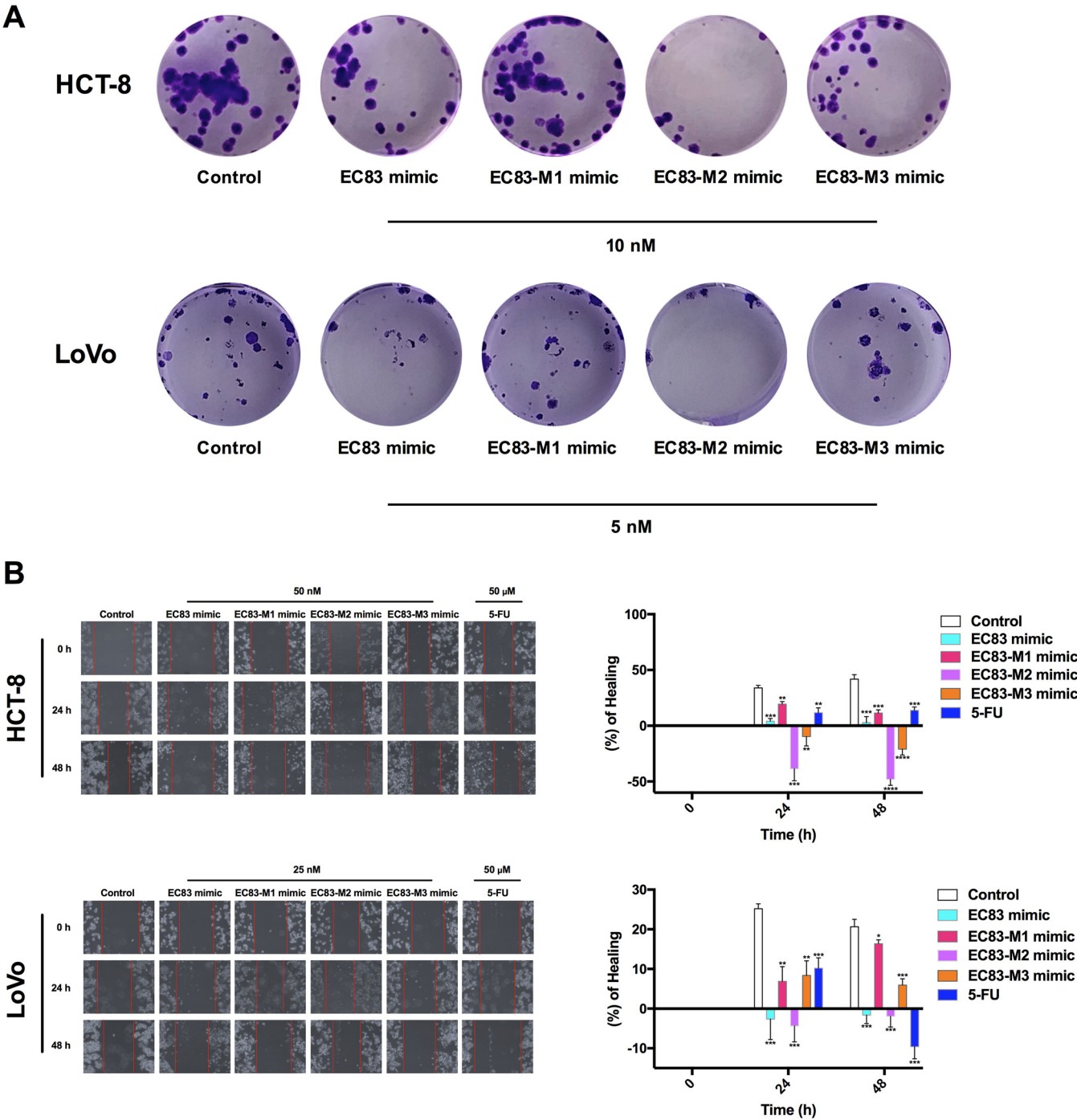

**FIG 5** Gm modification increases the inhibitory effects of EC83 mimics on colony formation and migration of CRC cells, while s⁴U does not. (A) Clonogenic assay of the EC83 mimic, the EC83-M1 mimic, the EC83-M2 mimic, and the EC83-M3 mimic on HCT-8 and LoVo cells. (B) Wound-healing assay of the EC83 mimic, the EC83-M1 mimic, the EC83-M2 mimic, and the EC83-M3 mimic on HCT-8 and LoVo cells. Data are shown as the means ± SD from three independent experiments. *, $P < 0.05$; **, $P < 0.01$; ***, $P < 0.001$; ****, $P < 0.0001$ (by two-tailed Student's *t* test).

root mean square deviation (RMSD) of the EC83 mimic is significantly larger than those of the modified mimics, indicating that the transformed helical structures (EC83-M1, -M2, and -M3 mimics) are more stable than the A-form standard helix (EC83 mimic).

To further evaluate the stability of the modified mimics, 200-ns simulations were carried out by increasing the temperature gradually from 300 K to 500 K (intervals of 10 K). The results showed that mimics would lose their stable structures in the order of EC83 mimic, EC83-M1 mimic or EC83-M3 mimic, and EC83-M2 mimic (Fig. 7D). Here, the unfolding structures were

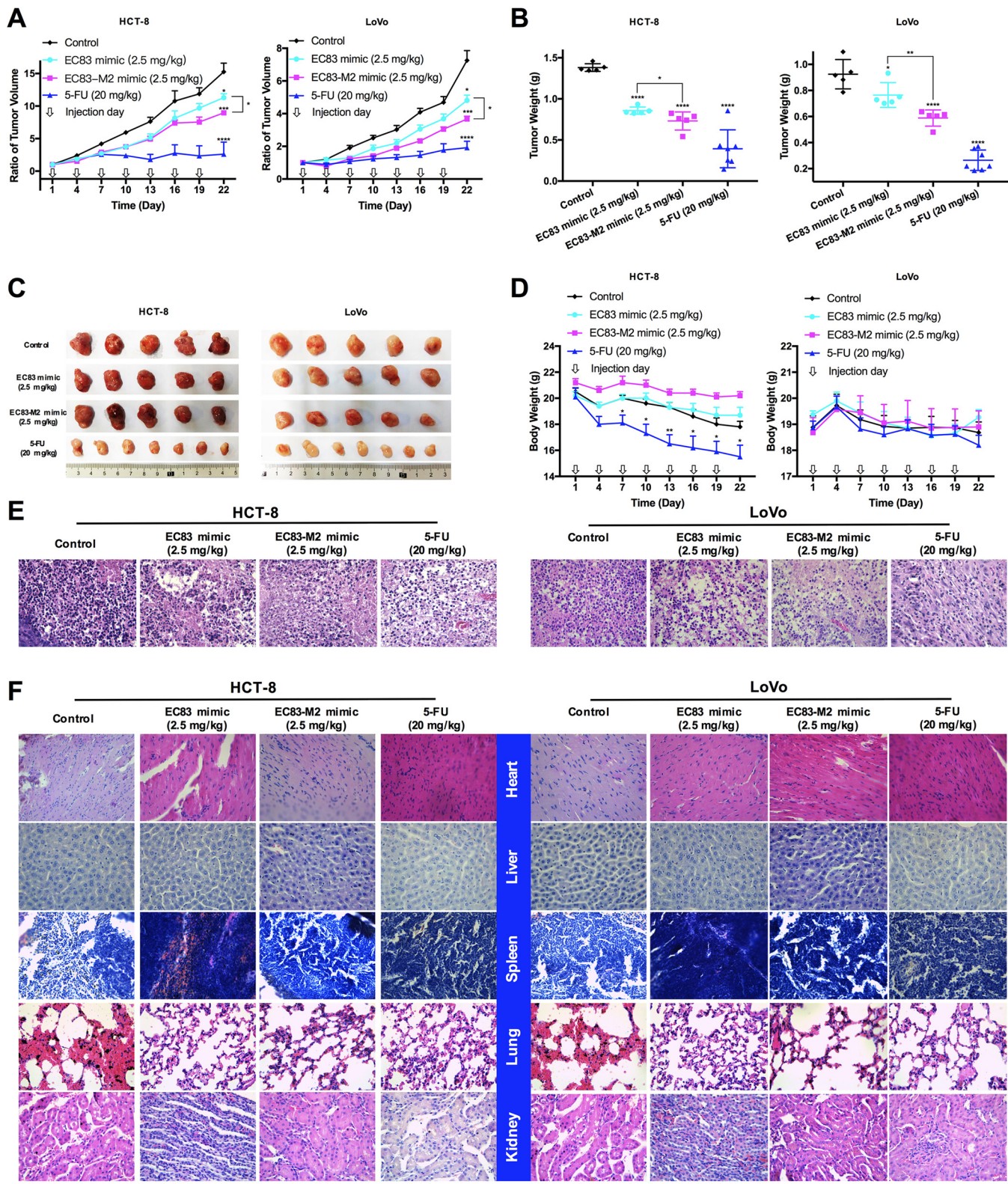

**FIG 6** Gm modification increases the inhibitory effects of EC83 mimics toward CRC tumor growth *in vivo*. (A) Growth rates of tumors with the encapsulated EC83 mimic and the EC83-M2 mimic, 5-FU, or HKP alone as a control. (B) Weights of tumors removed from the mice after day 22. (C) Images of each tumor in all groups. (D) Body weights of mice in all groups. (E and F) Representative images of hematoxylin and eosin-stained tumors (E) and major organs (F) of CRC xenograft-bearing nude mice in all groups. Data are shown as the means ± standard errors of the means (SEM). *, $P < 0.05$; ***, $P < 0.001$; ****, $P < 0.0001$ (by two-tailed Student's *t* test).

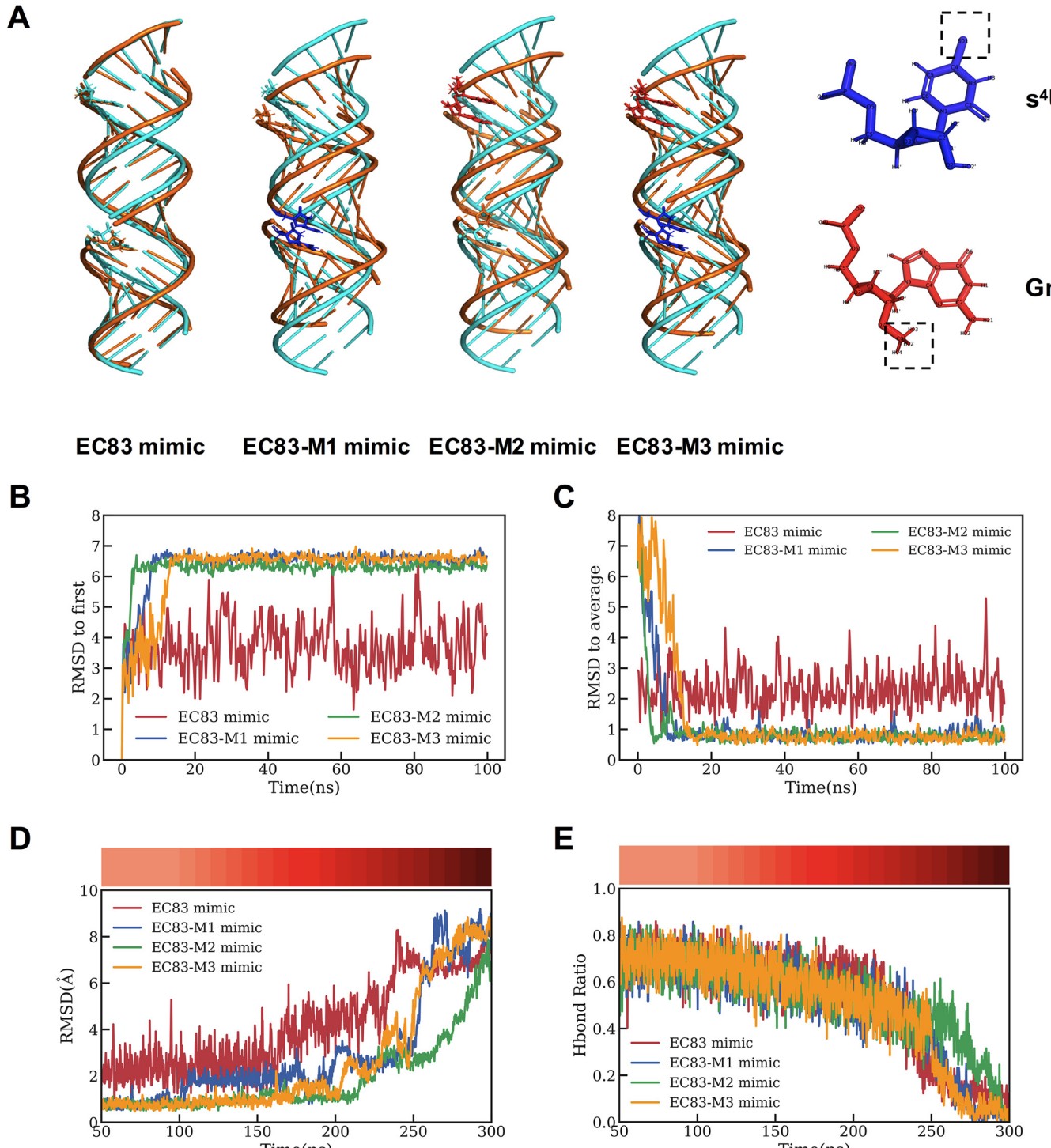

**FIG 7** Gm modification increases the tertiary structural stability of EC83 mimics. (A) Simulated three-dimensional (3D) initial structure (cyan) and transformed structure (gold) of the EC83 mimic and its modified derivatives. (B and C) RMSD for the initial A-form structure (B) and stable structures for four RNA mimics changing over time (C). (D and E) With the increasing temperature, the RMSD, using the stable structure as a reference (D), and the hydrogen bond (Hbond) ratios (E) of four RNA mimics change over time. The temperature at each time point is shown in the color bars at the top.

defined by their RMSD values above 5 Å relative to the stable structure. As further shown in Fig. 7E, the hydrogen bond (Hbond) ratio of the EC83-M2 mimic remained above 40% at 260 ns, while those of the other 3 were under 20%. These results demonstrated that the EC83-M2 mimic is structurally the most stable of the derivatives, which are also more stable than the unmodified EC83 mimic, suggesting that Gm modification enhances the tertiary

structural stability of EC83 mimics. On the other hand, $s^4U$ modification appears to not exhibit such a strong effect.

## DISCUSSION

CRC has become the third most frequently diagnosed malignancy worldwide (10.0%), accounting for ~576,858 deaths in 2020 (35). Risk factors of CRC include family history, a diet high in red and processed meat, heavy alcohol intake, smoking, being overweight or obese, age (being older), and inflammatory bowel disease (36). Currently, cytotoxic chemotherapy is widely used as a monotherapy or a component of combined therapy for CRC treatment (37). However, the development of resistance to treatment is a great challenge. RNAs are able to regulate gene expression and therefore may have the potential to be a new class of cancer therapeutics that are nontoxic and have desirable selectivity for a specific target gene (38). The present study describes, for the first time, the ability of tRNA halves and tRFs derived from NPECS that exhibit anticancer activity against colorectal cancer. They are very effective *in vitro*, with $IC_{50}$ values at the $10^{-8}$ M concentration range, compared to the $10^{-4}$ M range for 5-FU. Interestingly, tRFs exhibited higher potency than tRNA halves, suggesting that tRFs might play an important role in the cytotoxic effects of tRNAs derived from the gut microbiota. Importantly, the more potent EC83 mimic and its modified derivatives were observed to be just as effective on 5-FU-resistant CRC cells as on non-pathogenic cells, indicating that microbiota-derived tRFs may be an effective treatment even in CRC resistant to conventional chemotherapy. This observation supports the idea that the cytotoxic actions of tRFs differ from that of 5-FU. The exact mechanism of action remains to be investigated. It has been revealed that endogenous tRFs derived from human breast cancer cells could suppress breast cancer *in vitro* and *in vivo* by targeting YBX1 mRNA, suggesting that tRFs could target signaling pathways related to cell proliferation, thus enabling an effective intervention with the treatment of cancers (28). This, together with our previous study on a tRF derived from Chinese yew, which suppresses ovarian cancer by targeting the oncogene TRPA1 (31), raises the hypothesis that tRFs derived from non-pathogenic *E. coli* might also function via an RNA interference (RNAi) pathway. Further investigations on the molecular target of EC83 mimics should be performed via mRNA sequencing and experimental validations.

As a class of highly modified nucleic acids (39–44), tRFs have been proven to possess miRNA-like regulatory function in relation to RNA interference (33). Functional improvements have been reported following chemical modifications of tRFs (45). The loss of cytosine-5 RNA methylation was found to enhance the angiogenin-mediated endonucleolytic cleavage of tRNA, leading to an accumulation of 5′-tRFs and, subsequently, to a reduced cell size and increased apoptosis (46). Consistently, the present results revealed that Gm modification enhanced the cytotoxic effects of EC83 mimics perhaps via the observed increased stability of the tertiary structure. This was supported by the observation that $s^4U$ modification does not exhibit such a strong effect. Indeed, different RNAs have different potentials for encapsulation within lipid vesicles, which might be greatly impacted by their negative charges. However, the retention time values of the EC83 mimic and its derived modified mimics in our ion-pair chromatography analysis are very close (see Fig. S4 in the supplemental material). This indicated that the EC83 mimic and its derived modified mimics have similar negative charges, demonstrating that these RNAs have similar encapsulation rates within lipid vesicles. Overall, this first structure-activity investigation provided evidence that the bioactivity of tRF mimics can be impacted by chemical modifications. Further structural analysis is required to decipher the critical structural differences between these modified tRF mimics.

Recently, it was reported that colonic bacteria, especially PECSs, bear a close relationship with CRC development (47, 48). Thus, current interests in clinical studies focus on PECSs, such as enterohemorrhagic *E. coli* (EHEC) O157:H7, which produces metabolites or toxins that can cause DNA damage in the colon and CRC development (49–51). As the genome size and gene number of NPECSs (3.98 Mb and 3,696, respectively) differ from those of PECSs (5.86 Mb and 5,919) (52), NPECSs and PECSs may produce completely different profiles of tRNA fragments through different chemical modifications. This may raise the possibility of identifying CRC biomarkers from tRNA molecules derived from PECSs.

In summary, these findings revealed tRNA halves and especially tRFs as a new class of bioactive constituents derived from gut microorganisms themselves and provided new insights into their therapeutic effects on human diseases, which broadened our knowledge of the beneficial effects of gut microbiota. Also, the present study demonstrates that studies on biological functional molecules in the intestinal microbiota should not neglect tRFs since they are bioactive constituents (28). Research on tRFs will play an important role in biological research on gut microorganisms, including bacterium-bacterium interactions, the gut-brain axis, and the gut-liver axis, etc. Furthermore, with the increasing interest in the identification of tRFs in bacteria (53, 54), the guidance on the rational design of tRF therapeutics provided in this study suggests that further investigations should pay more attention to these therapeutics from probiotics. The innovative drug research on tRFs as potent druggable RNA molecules derived from intestinal microorganisms will open a new area in biomedical sciences.

## MATERIALS AND METHODS

**Chemicals and reagents.** *Escherichia coli* MRE600 total tRNA was purchased from Roche (Switzerland). Biotin-labeled single-stranded DNA oligonucleotides were obtained from BGI, China. A low-range single-stranded RNA (ssRNA) ladder was purchased from New England BioLabs (USA). Diethylpyrocarbonate (DEPC)-treated water, S1 nuclease, and polyacrylamide containing a ratio of acrylamide/bis of 19:1 (wt/wt) were purchased from Thermo (USA). Triethylammonium acetate, hexafluoro-2-propanol, and fluorouracil (5-FU) were purchased from Sigma (USA). Deionized water was prepared using a Millipore MilliQ Plus system. All reagents used were of analytical grade.

**Affinity purification of tRNAs from an NPECS.** Affinity purification was performed according to methods described previously by Tsurui et al. (55), with minor modifications. Briefly, biotin-labeled single-strand DNA (antisense) oligonucleotides complementary to tRNA-Val(UAC) (5′-GCCGACCCCCTCCTTGTAAGGGAGGTG CTC-3′) or tRNA-Leu(CAG) (5′-ACGTCCGTAAGGACACTAACACCTGAAGCT-3′) were mixed with total tRNA from *E. coli* and then denatured at 95°C for 5 min. After incubation at a temperature 5°C lower than the melting temperature ($T_m$) of each DNA probe for 1.5 h, streptavidin magnetic beads (Beaverbio, China) were added, and the mixture was incubated for another 30 min. Subsequently, the biotinylated DNA/tRNA-coated beads were separated with a magnet and washed at 40°C. Finally, the magnetic beads were incubated in RNase-free water at 65°C for 5 min to release the immobilized tRNA with a probe and then centrifuged at 10,000 × $g$ for 1 min. The supernatants were analyzed immediately by ultrahigh-performance liquid chromatography (UHPLC).

**S1 nuclease hydrolysis.** S1 nuclease was added to purified tRNA samples (8 U enzyme/500 ng tRNA) with reaction buffer (2 $\mu$L), the final volume was adjusted to 20 $\mu$L, and the mixture was incubated at 25°C for 40 min. The reaction was stopped by the addition of an EDTA solution (0.5 M; 0.5 $\mu$L). The supernatant was collected by centrifugation at 10,000 × $g$ for 1 min for UHPLC-MS analysis.

**Urea-polyacrylamide gel electrophoresis.** All samples were separated by vertical slab gel electrophoresis (Mini-Protean Tetra system; Bio-Rad, USA) using a 15% urea-polyacrylamide gel. Samples were electrophoresed at 250 V for 1 h at room temperature (25°C) and stained with 1× SYBR gold nucleic acid gel stain (Thermo) in MilliQ Plus water for 30 min, followed by imaging using a Bio-Rad imaging system under UV light.

**UHPLC-MS.** UHPLC (Agilent 1290 system) was performed using a $C_{18}$ column (Acquity UPLC [ultraperformance liquid chromatography] OST, 2.1 × 100 mm, 1.7-$\mu$m internal diameter [i.d.]; Waters, USA) at 60°C with a diode array detector. UHPLC-MS was performed using an Agilent 1290 system (Agilent Technologies, USA), equipped with a vacuum degasser, a quaternary pump, an autosampler, a diode array detector, and an Agilent ultrahigh-definition 6545 quadrupole time of flight (Q-TOF) mass spectrometer. Separation was carried out on an Acquity UPLC OST $C_{18}$ column (2.1 × 100 mm, 1.7-$\mu$m i.d.; Waters, USA) at 60°C. tRNAs were separated by eluting the column at a flow rate of 0.2 mL/min with a mobile phase of 100 mM 1,1,1,3,3,3-Hexafluoro-2-propanol (HFIP) plus 15 mM triethylammonium acetate (TEAA) containing methanol (MeOH) at the following concentrations: 1% (vol/vol) for 1.5 min, 1 to 14% over 1.5 to 8.3 min, and finally 14 to 17% over 8.3 to 16.5 min. Electrospray ionization (ESI) conditions were as follows: gas temperature of 320°C, spray voltage of 3.5 kV, and sheath gas flow and temperature set at 12 L/min and 350°C, respectively. Fractions corresponding to each chromatographic peak were collected and freeze-dried. For MS experiments, samples were analyzed in negative mode over an $m/z$ range of 500 to 3,200.

**Cell culture.** The human ovarian cancer cell line A2780 was purchased from KeyGen Biotech (China). The human liver cancer cell line HepG2, the human breast cancer cell line MDA-MB-231, the human colorectal cancer cell lines HCT-8 and LoVo along with their 5-FU-resistant strains (HCT-8/5-FU and LoVo/5-FU), and the human colon epithelial cell line HCoEpiC were purchased from the American Type Culture Collection (ATCC). A2780, LoVo/5-FU, HCT-8, HCT-8/5-FU, and HCoEpiC cells were cultured in RPMI 1640 medium (Gibco, New Zealand). LoVo cells were cultured in F-12K medium (Thermo). HepG2 cells were cultured in minimal essential medium (MEM) (Thermo). MDA-MB-231 cells were cultured in Leibovitz's L-15 medium (Thermo). All culture media contained 10% fetal bovine serum (FBS) and 1% penicillin/streptomycin (P/S). All cell lines were cultured in a humidified 5% $CO_2$ atmosphere at 37°C, except for MDA-MB-231 cells, which were cultured in a humidified 100% air atmosphere at 37°C. All tested RNA samples were dissolved in nuclease-free water and stored at −80°C before use. 5-FU was dissolved in dimethyl sulfoxide (DMSO) and used as a positive control.

**Cytotoxicity determination.** Cells (5 × $10^3$ in 100 $\mu$L culture medium) were seeded onto 96-well plates. After 20 h, cells were treated with various concentrations of RNA sample solutions with Lipofectamine RNAiMAX transfection reagent in Opti-MEM medium (Thermo) according to the manufacturer's instructions.

Cells without any treatment were used as controls, and cells treated with liposomes were used as treatment controls. Cell viability was determined after 48 h. An MTT [3-(4,5-dimethylthiazol-2-yl)-2,5-diphenyltetrazolium bromide] solution (50 $\mu$L per well; 1 mg/mL) (Thermo) was added to each well; the mixture was incubated for 4 h at 37°C, followed by DMSO (200 $\mu$L); and the $A_{570}$ was measured using a SpectraMax 190 microplate reader (Molecular Devices, USA). Dose-response curves were constructed and the $IC_{50}$ values were calculated using GraphPad Prism 5.0 (GraphPad, USA). Each experiment was carried out three times. $IC_{50}$ results were expressed as means $\pm$ standard deviations (SD).

**Clonogenic assay.** The clonogenic assay was performed according to methods described previously by Franken et al. (56), with minor modifications. Briefly, cells were plated at a density of 1,000 cells/well with culture medium in 6-well plates. After 20 h, the medium was changed to medium containing RNA samples (10 nM for HCT-8 and 5 nM for LoVo cells) with liposomal transfection or blank Opti-MEM for a further 48 h. Cells were maintained in normal culture medium for the following 14 days. After fixation for 20 min with a 4% paraformaldehyde (PFA) fix solution (Beyotime, China), the cells were stained with crystal violet (Beyotime, China) for 10 min. Finally, the numbers of colonies with more than 50 individual cells were counted using ImageJ software.

**Wound-healing assay.** Cells ($5 \times 10^5$ in 100 $\mu$L culture medium) were grown in 6-well plates for 20 h until confluent. A scratch was made by using a sterile 1-mL pipette tip, and the medium was changed to medium containing RNA samples (50 nM for HCT-8 and 25 nM for LoVo cells) with liposomal transfection or 5-FU at 50 $\mu$M. The cells were viewed with a 10× objective and photographed using a phase-contrast microscope (Leica Microsystems, Germany) at various time points (0, 24, and 48 h). ImageJ software was applied to quantify the area of the wound created. The wound-healing rate was calculated using the following formula: wound-healing rate = [(wound area at 0 h − wound area at 24 or 48 h)/wound area at 0 h] × 100.

**Animal study.** Animal care and use protocols were approved by the Ethics Committee of the Macau University of Science and Technology. HCT-8 cells ($2.0 \times 10^6$) and LoVo cells ($1.0 \times 10^7$) were injected subcutaneously under the armpits of 8-week-old BALB/c female nude mice (Shanghai SLAC Laboratory Animal Co., Ltd., China). When the tumors reached 100 mm³, the EC83 mimic and EC83-M2 mimic encapsulated in HKP nanoparticles were administered by intratumoral injection (2.5 mg/kg of body weight) once every 3 days. 5-FU (20 mg/kg) and histidine-lysine polymer (HKP) were used as positive and negative controls, respectively. The animals were sacrificed on day 22. Tumor diameters were measured at the points of maximum length and maximum width with digital calipers. Tumor volumes were calculated by the following formula: volume = (width)² × length/2. Data were statistically analyzed using GraphPad Prism 5.0.

**Histological analysis.** The tissues from nude mice were fixed in 4% PFA and embedded in paraffin. Three-micrometer to 5-$\mu$m paraffin sections were cut and stained with hematoxylin and eosin (H&E) to determine the efficacy and side effects of the EC83 mimic and the EC83-M2 mimic.

**Computational simulations of tertiary structures.** The initial standard A-form double-helix structures for the EC83 mimic and its modified derivatives were built by NAB (57) in the Amber18 package (58). The force field used was OL3 (59) (leaprc.RNA.OL3 in Amber18), and the parameters for the modified nucleotides (s⁴U and Gm) were described previously by Aduri et al. (60). Each system was solvated in a 12-Å transferable interatomic potential with three points model (TIP3P) cubic water box (~6,000 water molecules), and 42 Na⁺ were then randomly added to neutralize the whole system. To minimize the systems before simulation, RNA molecules were first restricted by 20 kcal/mol Å harmonic potential with the energy of the solvent minimized for 8,000 steps, where the first 4,000 were steepest-descent steps and the last 4,000 were conjugated-gradient steps. Furthermore, another 8,000 minimization steps were performed in the same way except that the constraints exerted on the solvate were removed.

After energy optimization, the system was heated from 0 to 300 K, and temperature equilibration was allowed at 300 K for 200 ps. Subsequently, another 200-ps equilibration simulation was performed. The solvate was restricted as described above. Besides, the time step was set to 1 fs and run at NPT ensemble using the parallel version of pmemd (61), with the primary molecular dynamics engine within Amber. A 300-ns production simulation was performed for each RNA molecule. The temperature was equilibrated at 300 K by Langevin dynamics in the first 100 ns and then gradually increased to 500 K in the last 200 ns with 10 K intervals. The time step was increased from 1 to 2 fs due to the application of the SHAKE algorithm. The nonbond cutoff was set to 10 Å, and the long-range electrostatic interactions were evaluated by the pattern method extension (PME) method. The pressure was kept at around 10⁵ Pa (0.987 atm), and all simulations were performed on Nvidia graphic processing unit (GPU) cards by the GPU version of pmemd.

The stabilities of the RNA molecules were compared in terms of backbone (P, O3′, O5′, C3′, C4′, and C5′) RMSDs based on reference to the stable structures and the extent of hydrogen bonding in 22 bp. As the fluctuation of the RMSD for each RNA molecule was small, their stable structures were the average structures in 50 to 100 ns. Furthermore, the two criteria were calculated by cpptraj (62) in Amber, and the following data analysis was based on numpy and matplotlib, which are two popular python packages. The tertiary structures were visualized using PyMOL (63).

**Statistical and data analyses.** All experimental results were expressed as means $\pm$ SD. Statistical significance was analyzed using two-tailed Student's *t* test (GraphPad Prism) or one-way analysis of variance (ANOVA) followed by *post hoc* analysis.

## SUPPLEMENTAL MATERIAL

Supplemental material is available online only.

**FIG S1**, TIF file, 0.9 MB.

**FIG S2**, TIF file, 0.9 MB.

**FIG S3**, TIF file, 2.1 MB.

FIG S4, TIF file, 0.5 MB.
TABLE S1, DOCX file, 0.02 MB.
TABLE S2, DOCX file, 0.02 MB.

## ACKNOWLEDGMENTS

This work was financially funded by the Science and Technology Development Fund, Macau SAR (file no. 0023/2019/AKP, 015/2017/AFJ), and the State Key Laboratory of Chemical Oncogenomics.

We declare that there are no conflicts of interest.

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
