## [Reviewer comments · mSystems]

The antitumor activity of tRNA-derived fragments and tRNA halves from non-pathogenic *Escherichia coli* strain on colorectal cancer and their structure-activity relationship

Kai-Yue Cao, Yu Pan, Tong-Meng Yan, Peng Tao, Yi Xiao, and Zhi-Hong Jiang

Corresponding Author(s): Zhi-Hong Jiang, Macau University of Science and Technology

Review Timeline:

Submission Date:	February 23, 2022
Editorial Decision:	March 18, 2022
Revision Received:	March 20, 2022
Accepted:	March 23, 2022

Editor: Zackery Bulman

Reviewer(s): The reviewers have opted to remain anonymous.

Transaction Report:

DOI: <https://doi.org/10.1128/msystems.00164-22>

March 18, 2022

Prof. Zhi-Hong Jiang
Macau University of Science and Technology
Macao
Macau

Re: mSystems00164-22 (The antitumor activity of tRNA-derived fragments and tRNA halves from non-pathogenic Escherichia coli strain on colorectal cancer and their structure-activity relationship)

Dear Prof. Zhi-Hong Jiang:

Thank you for submitting your manuscript to mSystems. We have completed our review and I am pleased to inform you that, in principle, we expect to accept it for publication in mSystems. However, acceptance will not be final until you have adequately addressed the reviewer comments. There are a few minor issues, which I agree with, that have been identified by the reviewer.

Preparing Revision Guidelines

Sincerely,

Zackery Bulman

Editor, mSystems

Journals Department
Reviewer comments:

Reviewer (Comments for the Author):

The MS entitled 'The antitumor activity of tRNA-derived fragments and tRNA halves from non-pathogenic Escherichia coli strain on colorectal cancer and their structure-activity relationship' by Cao et al is rebuttal of work presented to review in June 2020. The MS includes a new mouse experiment which is consistent with the previous data and demonstrates the role of tRSs in vivo. The text has been significantly improved. The quality of the MS for cancer biology is rather limited, but the work is an interesting addition to existing knowledge on tRNA fragments (tRFs) and their implication into cell biology. The authors demonstrated that tRfs from non -pathogenic E.coli are cytotoxic to cell lines and that they present an anti-tumor role in vivo.

Specific comments:

- Please update the statistics for cancers: there is GLOBOCAN from 2021 (<https://acsjournals.onlinelibrary.wiley.com/doi/full/10.3322/caac.21660>).
- Figure 5A. What was the reason for choosing concentration: 10 nM and 5 nM instead of IC50?
- Figure 5B. The diagram: How do the Authors interpret the negative values? When the cells do not move forward it usually means that they are dead, but then the „y value is usually 0 not below 0.
- Could the authors speculate in the discussion what is the proposed signalling pathway?

Response to Reviewer's Comments

Reviewer (Comments for the Author):

Point 1. Please update the statistics for cancers: there is GLOBOCAN from 2021 (<https://acsjournals.onlinelibrary.wiley.com/doi/full/10.3322/caac.21660>).

Response 1: Thanks for the reviewer's suggestion. The above literature has been added as Reference 35 cited in Line 207-208, Page 12.

Point 2. Figure 5A. What was the reason for choosing concentration: 10 nM and 5 nM instead of IC₅₀?

Response 2: Thanks for the reviewer's comment. In the present study, the purpose of clonogenic assay is to evaluate the inhibitory effect of tRF mimics derived from non-pathogenic *E. coli* on the colony formation of CRC cells. Thus, the RNA dosage should not be too high to cause significant cell death during the long-time cell culture. Thus, the dosages of 10 nM and 5 nM were selected in this study instead of IC₅₀ since all tRF mimics exhibited no significant cytotoxicity towards CRC cells under these concentrations.

Point 3. Figure 5B. The diagram: How do the Authors interpret the negative values? When the cells do not move forward it usually means that they are dead, but then the „y value is usually 0 not below 0.

Response 3: Thanks for the reviewer's comment. We explained in the manuscript that “Wound healing rate = [(Wound area at 0 h – Wound area at 24 or 48 h)/Wound area at 0 h] × 100”. Since the RNA dosage of 50 nM and 25 nM induce cell deaths, the wound healing rate of some tRF mimics are negative because their wound areas at 24 or 48 h are larger than those at 0 h.

Point 4. Could the authors speculate in the discussion what is the proposed signaling pathway?

Response 4: Thanks for the reviewer's comments. It has been revealed that the endogenous tRFs derived from human breast cancer cells could suppress breast cancer *in vitro* and *in vivo* by targeting oncogene YBX1 gene, suggesting that tRFs could target the signaling pathways related to cell proliferation, thus enable an effective

intervention with the treatment of cancers (1). This, together with our previous study on tRF derived from Chinese yew which suppresses ovarian cancer *via* targeting oncogene TRPA1 (2), raise the hypothesis that tRF derived from non-pathogenic *E. coli* might also function *via* RNAi pathway. Further investigations on molecular target of EC83 mimic should be performed *via* mRNA sequencing and experimental validations. The above discussions were added in Line 225-233, Page 12-13.

References

1. Goodarzi H, Liu XH, Nguyen HCB, Zhang S, Fish L, Tavazoie SF. 2015. Endogenous tRNA-derived fragments suppress breast cancer progression via YBX1 displacement. *Cell* 161:790–802.
2. Cao KY, Yan TM, Zhang JZ, Chan TF, Li J, Li C, Elaine Lai HL, Gao J, Zhang BX, Jiang ZH. 2022. A tRNA-derived fragment from Chinese yew suppresses ovarian cancer growth via targeting TRPA1. *Mol Ther-Nucl Acids* 27:718-732.

March 23, 2022

Prof. Zhi-Hong Jiang
Macau University of Science and Technology
Macao
Macau

Re: mSystems00164-22R1 (The antitumor activity of tRNA-derived fragments and tRNA halves from non-pathogenic Escherichia coli strain on colorectal cancer and their structure-activity relationship)

Dear Prof. Zhi-Hong Jiang:

Thank you for revising and resubmitting your manuscript. I am pleased to inform you that your manuscript has been accepted, and I am forwarding it to the ASM Journals Department for publication. For your reference, ASM Journals' address is given below. Before it can be scheduled for publication, your manuscript will be checked by the mSystems production staff to make sure that all elements meet the technical requirements for publication. They will contact you if anything needs to be revised before copyediting and production can begin. Otherwise, you will be notified when your proofs are ready to be viewed.

Publication Fees:

We recognize that the video files can become quite large, and so to avoid quality loss ASM suggests sending the video file via <https://www.wetransfer.com/>. When you have a final version of the video and the still ready to share, please send it to mSystems staff at mssystems@asmusa.org.

For mSystems research articles, if you would like to submit an image for consideration as the Featured Image for an issue, please contact mSystems staff at mssystems@asmusa.org.

Sincerely,

Zackery Bulman
Editor, mSystems

Journals Department
Table S2: Accept
Fig. S2: Accept
Fig. S3: Accept
Fig. S4: Accept
Table S1: Accept
Fig. S1: Accept